# Survival in a consecutive series of 467 glioblastoma patients: Association with prognostic factors and treatment at recurrence at two independent institutions

Hanne Blakstad[1,2☯], Jorunn Brekke[3☯], Mohummad Aminur Rahman[3,4], Victoria Smith Arnesen[3,4], Hrvoje Miletic[4,5], Petter Brandal[1,6], Stein Atle Lie[7], Martha Chekenya[4‡*], Dorota Goplen[3‡]

1 Department of Oncology, The Norwegian Radium Hospital, Oslo University Hospital, Oslo, Norway, 2 Institute for Clinical Medicine, University of Oslo, Oslo, Norway, 3 Department of Oncology and Medical Physics, Haukeland University Hospital, Bergen, Norway, 4 Institute for Biomedicine, University of Bergen, Bergen, Norway, 5 Department of Pathology, Haukeland University Hospital, Bergen, Norway, 6 Institute for Cancer Genetics and Informatics, Oslo University Hospital, Oslo, Norway, 7 Institute for Clinical Dentistry, University of Bergen, Bergen, Norway

☯ These authors contributed equally to this work.
‡ MC and DG are co-senior authors on this work.
* martha.chekenya@biomed.uib.no

**Data Availability Statement:** Data cannot be shared publicly because of this is sensitive patient

## Abstract

Therapy of recurrent glioblastoma (GBM) is challenging due to lack of standard treatment. We investigated physicians' treatment choice at recurrence and prognostic and predictive factors for survival in GBM patients from Norway's two largest regional hospitals. Clinico-pathological data from n = 467 patients treated at Haukeland and Oslo university hospitals from January 2015 to December 2017 was collected. Data included tumour location, pro-moter methylation of $O^6$ methylguanine-DNA methyltransferase (*MGMT*) and mutation of isocitrate dehydrogenase (*IDH*), patient age, sex, extent of resection at primary diagnosis and treatment at successive tumour recurrences. Cox-proportional hazards regression adjusting for multiple risk factors was used. Median overall survival (OS) was 12.1 months and 21.4% and 6.8% of patients were alive at 2 and 5 years, respectively. Median progression-free survival was 8.1 months. Treatment at recurrence varied but was not associated with difference in overall survival (OS) ($p = 0.201$). Age, *MGMT* hypermethylation, tumour location and extent of resection were independent prognostic factors. Patients who received 60 Gray radiotherapy with concomitant and adjuvant temozolomide at primary diagnosis had 16.1 months median OS and 9.3% were alive at 5 years. Patients eligible for gamma knife/stereotactic radiosurgery alone or combined with chemotherapy at first recurrence had superior survival compared to chemotherapy alone ($p<0.001$). At second recurrence, combination chemotherapy with or without bevacizumab were both superior to no treatment. Treatment at recurrence differed between the institutions but there was no difference in median OS, indicating that it is the disease biology that dictates patient outcome.

data. Data are available from the corresponding authors (MC) as they are approved by the Ethics Committee (approval number 2017/2084) for researchers who meet the criteria for access to confidential data. Non-author institutional point of contact person to fill data access queries: Berit Bølge Tysnes, PhD Department of Biomedicine University of Bergen Jonas Lies vei 91 5009 Bergen, Norway Email: Berit.Tysnes@uib.no Phone: +4790778791 / +4755586093.

**Funding:** 1. MC Grant # 190170 Norwegian Cancer Society https://kreftforeningen.no/ Sponsors or funders did not play any role in the study design, data collection and analysis, decision to publish, or preparation of the manuscript 2. MC and DG Program for klinisk behandlingsforskning - KLINBEFORSK https://kliniskforskning.rhf-forsk. org/ Sponsors or funders did not play any role in the study design, data collection and analysis, decision to publish, or preparation of the manuscript.

**Competing interests:** The authors have declared that no competing interests exist.

## Introduction

Glioblastoma (GBM) is the most malignant primary brain tumour in adults [1, 2]. The incidence of GBM in the developed world is approximately 3.2/100.000 people [2]. In Norway, it has been 4.2–4.9/100.000 people (*personal communication;* Edrun Andrea Schnell, 2016 [3]). Current standard treatment consists of maximal safe resection, followed by concomitant chemoradiotherapy with temozolomide (TMZ) 75 mg/m$^2$ with irradiation 60 Gray (Gy)/30 fractions, and 6 cycles of adjuvant TMZ [4]. This multimodal treatment is most suitable for fit patients under 70 years of age, fosters a progression-free survival (PFS) of ~6.9 months, and extends overall survival (OS) to 14.6 months [5]. Silencing O$^6$ methylguanine-DNA methyltransferase (*MGMT*) by promoter hypermethylation is both prognostic and predictive of response to alkylating temozolomide chemotherapy [6].

There is currently no standard treatment for recurrent GBM, the chemotherapy options are limited and the European Association of Neuro-Oncology (EANO) recommends the inclusion of these patients in clinical trials [7, 8]. If a single-arm study is chosen, it is important to have a valid comparator. A good comparator is a matched historical patient cohort treated at the institutions where a clinical trial is conducted, including patients from independent clinical centres. Knowledge of patient survival at different institutions is valuable from a population-based perspective.

We present retrospectively analysed clinicopathological glioblastoma data from two independent tertiary referral centres in Norway. Approximately 155 GBM patients are treated every year within the eastern and western regional Norwegian health authorities with a combined population of 4.1 million people. We aimed to evaluate survival for a consecutive group of patients diagnosed with histologically confirmed GBM in these two healthcare regions, to find out whether differences in treatment practice of recurrent GBM impacted survival.

## Materials and methods

Several methods are detailed in S1 File.

### Study design

This is a population-based retrospective study.

### Patient cohort

A consecutive series of 467 retrospective patients with histologically confirmed GBM diagnosed and treated at Oslo University Hospital (OUH) (n = 327) and at Haukeland University Hospital (HUH) (n = 140) from January 2015 to December 2017 were included in the study cohort. At this time point, the diagnosis of GBM was according to the 4th edition of the WHO classification of tumors of the central nervous system [9]. The time point for data inclusion was chosen because the majority of patients had known isocitrate dehydrogenase (*IDH*) and *MGMT* status and it allowed an adequate follow-up period after the standard Stupp regimen [5]. All patients were identified using the hospitals' internal quality registry. Patients with secondary GBM following a previously known lower-grade glioma and patients with GBM as second primary were included, making the cohort consecutive and complete.

### Statistical analysis

Patient OS was calculated from date of primary surgery to date of death or date of the administrative end of study (April 1$^{st}$, 2020), which was considered censored observations. PFS was

calculated from date of primary surgery to date of tumour progression, death, or date of censoring, whichever occurred first. Inverse Kaplan-Meier was used for calculating median follow-up [10] and patients who were alive at the last follow-up were censored from survival analyses. Patients who died before the first, second or third MRI-confirmed progression were considered to experience tumour progression at the time of death. To analyse the impact of the previous treatment on the recurrent treatments, analysis using time-dependent covariates was performed. The Kaplan–Meier method with log-rank test [11] was used for survival probabilities. Cox proportional hazards regression with pairwise comparison, adjusted for multiple testing using Scheffé's posthoc test, was used to analyse the effect of multiple risk factors on mortality. Stata version 16.1 (StataCorp LLC, Texas, USA) was used for statistical analyses. Two-sided $P$-values less than 0.05 were considered significant. Descriptive statistics were reported as frequencies unless otherwise stated.

### Ethics

Regional Committee for Medical and Research Ethics for Western Norway (REC West) approved the retrospective patient identification and collection of clinicopathological data (2017/2084). Exemption from the need to obtain informed consent from included patients, including the few surviving patients at the time of data collection (n = 9 and n = 43; HUH and OUH, respectively), was granted by REC West.

## Results

Median overall survival for the 467 patients was 12.1 months (S1A Fig), where 50.3%, 95% CI [0.46–0.55] were alive at one year, 21.4%, 95% CI [0.18–0.25] at 2 years and 6.8%, 95% CI [0.04–0.11] at 5 years, Fig 1A. The median follow-up time was 42.5 months and 415 patients were deceased by the time of analysis.

### Patient and tumour characteristics

Baseline patient and tumour characteristics are presented in Table 1. There were 273 (58.5%) males and 194 (41.5%) females, with a mean age of 61.8 ±12.2 years (range 17–85). The majority of tumours were *IDH* wild type (74.7%) and *MGMT* promoter unmethylated (43.7%), whereas 17.3% had unknown *IDH* status and 17.1% had unknown *MGMT* promoter methylation status. In 45.4% of patients (n = 212) tumours were located within the right hemisphere, in 39.2% (n = 183) it was located in the left hemisphere, and in 15.4% (n = 72) tumours were situated in the midline or both hemispheres. Approximately 16% (n = 76) had multifocal neoplastic disease. The majority of patients (n = 438, 93.8%) had primary GBM, 27 patients (5.8%) had secondary GBM, and 2 patients (0.4%) had GBM as second primary neoplasm.

### Age, *MGMT* promoter methylation, tumour location, and extent of surgical resection were independently prognostic for patient outcome

Increasing age with each advancing decade was prognostic for patients' outcomes, $HR_{1.34}$, 95% CI [1.22–1.46], Log rank$_{44.17}$, $p<0.001$. Patients younger than 60 years had significantly better prognosis compared to patients 60–69 years old (median OS 16.3 *vs*. 12.1 months), $HR_{1.43}$, 95% CI [1.13–1.80], $p = 0.003$; and to those ≥70 years, (median OS 16.3 *vs*. 8.6 months), $HR_{2.32}$, 95% CI [1.83–2.95], $p<0.001$, Fig 2A and Table 1. Sex was not prognostic, Table 1.

Patients with tumours harbouring hypermethylated *MGMT* promoter had a median OS of 18.2 months compared to 11.4 months of *MGMT* unmethylated patients, $HR_{2.10}$, 95% CI [1.68–2.62], $p<0.001$, Fig 2B. Patients with unknown *MGMT* status had poorer median OS at

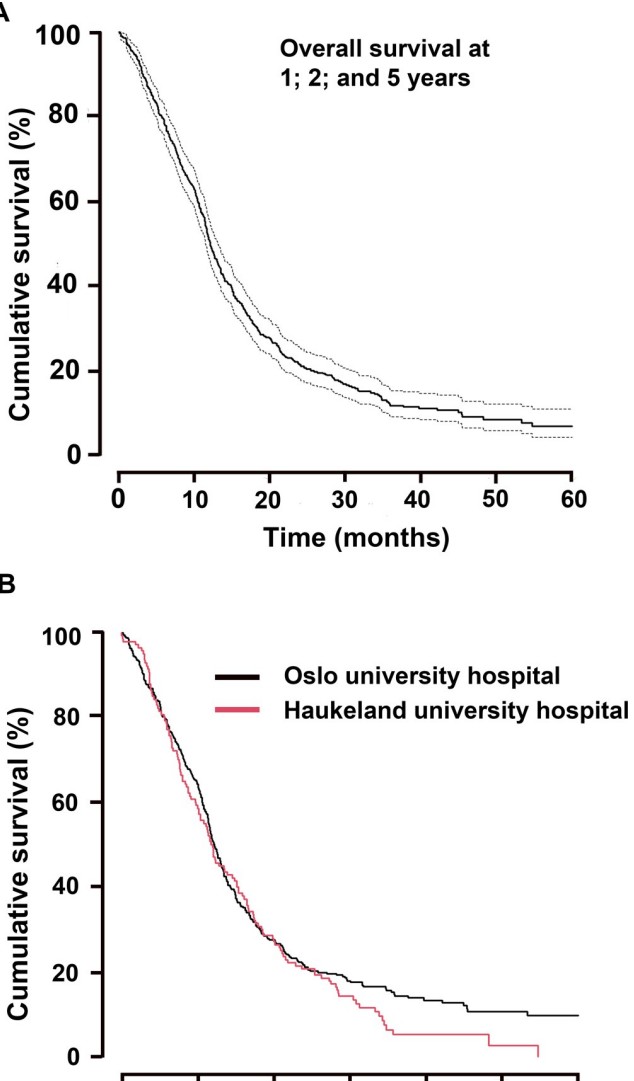

**Fig 1. Overall survival of patient population.** (A) Cumulative overall (%) survival time in months and 95% confidence intervals after 1-, 2 -and 5-year follow-up, total n = 467 GBM patients, (B) Cumulative overall (%) survival time in months and 95% confidence intervals after 1 -, 2- and 5-year follow-up at Oslo University Hospital (OUH) and Haukeland University Hospital (HUH).

8.7 months compared to hypermethylated patients, $HR_{2.23}$, 95% CI [1.68–2.95], $p<0.001$, Fig 2B and Table 1.

   *IDH*-mutation was not an independent prognostic factor for survival probability in multivariate analysis including all patients ($p = 0.057$), Table 1. If excluding all secondary GBM and second primary GBM, however, *IDH*-mutation status came across as an independent significant factor for better overall survival ($p = 0.006$, $HR_{0.43}$, 95% CI [0.24, 0.78]). Patients with unknown *IDH* mutational status in their tumours had significantly shorter median OS 9.5 months compared to those with *IDH* wild type tumours, $HR_{1.36}$, 95% CI [1.06–1.76], $p = 0.016$, Fig 2C and Table 1.

**Table 1.  Baseline patient and tumour characteristics and their association with survival.**

| Characteristics | Total n = 467 (%) | Overall survival (OS) | | | Median OS | Unadjusted analyses | | Adjusted analyses | |
|---|---|---|---|---|---|---|---|---|---|
| | | 1 year | 2 year | 5 year | months | Hazard ratio (95% CI) | P-value | Hazard ratio (95% CI) | P-value |
| **Sex** | | | | | | | | | |
| Male | 273 (58.5) | 50.6% | 22.3% | 5.9% | 12.1 | 1 | - | | |
| Female | 194 (41.5) | 50.0% | 20.1% | 8.5% | 12.0 | 0.97 (0.80, 1.18) | 0.746 | | |
| **Age (years)** | | | | | | | | | |
| <60 | 187 (40.0) | 63.1% | 31.6% | 11.6% | 16.3 | 1 | - | 1 | - |
| 60–69 | 144 (30.8) | 52.1% | 22.9% | 4.6% | 12.1 | 1.43 (1.13, 1.80) | **0.003** | 1.51 (1.19, 1.92) | **0.001** |
| ≥70 | 136 (29.1) | 30.9% | 5.9% | 3.9% | 8.6 | 2.32 (1.83, 2.95) | **<0.001** | 2.39 (1.86, 3.06) | **<0.001** |
| **Tumour location** | | | | | | | | | |
| Right side | 212 (45.4) | 50.5% | 20.3% | 2.1% | 12.1 | 1 | - | 1 | - |
| Left side | 183 (39.2) | 61.2% | 27.3% | 14.6% | 15.0 | 0.73 (0.59, 0.91) | **0.005** | 0.71 (0.58, 0.88) | **0.002** |
| Midline/bilateral | 72 (15.4) | 22.2% | 9.7% | 4.6% | 6.0 | 1.91 (1.45, 2.52) | **<0.001** | 1.22 (0.81, 1.82) | 0.339 |
| **Multifocality** | | | | | | | | | |
| Solitary | 391 (83.7) | 56.0% | 24.6% | 7.9% | 13.3 | 1 | - | 1 | |
| Multifocal | 76 (16.3) | 21.1% | 5.3% | 0% | 6.2 | 2.67 (2.06, 3.46) | **<0.001** | 1.57 (1.04, 2.37) | **0.034** |
| **Surgical resection** | | | | | | | | | |
| GTR | 168 (36.0) | 70.2% | 33.9% | 11.6% | 17.2 | 1 | - | 1 | - |
| STR | 221 (47.3) | 46.2% | 18.1% | 6.2% | 11.4 | 1.75 (1.40, 2.17) | **<0.001** | 1.62 (1.29, 2.04) | **<0.001** |
| Biopsy | 78 (16.7) | 19.2% | 3.9% | 0% | 6.8 | 3.58 (2.69, 4.76) | **<0.001** | 2.39 (1.70, 3.37) | **<0.001** |
| ***MGMT* promoter status** | | | | | | | | | |
| Hypermethylated | 183 (39.2) | 65.6% | 38.8% | 12.8% | 18.2 | 1 | - | 1 | |
| Unmethylated | 204 (43.7) | 45.6% | 9.8% | 0% | 11.4 | 2.10 (1.68, 2.62) | **<0.001** | 2.09 (1.66, 2.63) | **<0.001** |
| Unknown | 80 (17.1) | 27.5% | 11.3% | 5.8% | 8.7 | 2.23 (1.68, 2.95) | **<0.001** | 1.67 (0.88, 3.15) | 0.116 |
| ***IDH* status** | | | | | | | | | |
| Mutated | 37 (7.9) | 70.3% | 54.1% | 12.2% | 28.2 | 0.46 (0.31, 0.70) | **<0.001** | 0.65 (0.42, 1.01) | 0.057 |
| Wild type | 349 (74.7) | 53.3% | 20.1% | 5.5% | 12.5 | 1 | - | 1 | - |
| Unknown | 81 (17.3) | 28.4% | 12.4% | 7.2% | 9.5 | 1.36 (1.06, 1.76) | **0.016** | 0.83 (0.44, 1.55) | 0.554 |
| **Histology** | | | | | | | | | |
| Primary GBM | 438 (93.8) | 50.2% | 21.7% | 7.0% | 12.1 | 1 | - | | |
| Secondary GBM* | 27 (5.8) | 55.6% | 18.5% | 6.9% | 14.4 | 0.89 (0.59, 1.35) | 0.586 | | |
| Second primary GBM* | 2 (0.4) | 0% | 0% | 0% | 1.4 | 5.74 (1.42, 23.24) | **0.014** | | |

Significant *p*-values are highlighted in bold.

*Survival times calculated from time of GBM diagnosis. 51.7% of these patients received chemotherapy only as primary treatment for GBM because they had received radiation therapy earlier.

Abbreviations: CI: confidence interval; GTR: gross total resection; STR: subtotal resection; *IDH*: isocitrate dehydrogenase; *MGMT*: O$^6$ methylguanine-DNA methyltransferase; GBM: glioblastoma

## GBM location in the left hemisphere was independently associated with a reduced risk of death

Tumour location within the left, as opposed to the right hemisphere, was associated with reduced risk of death, (median OS 15 *vs.* 12.1 months), $HR_{0.73}$, 95% CI [0.59–0.91], *p* = 0.005, Fig 2D and Table 1. In adjusted analyses right-sided tumour was associated with increased risk of death $HR_{1.39}$, 95% CI [1.12–1.72], *p* = 0.003, as was multifocality (median OS 6.2 months), $HR_{2.46}$, 95% CI [1.65–3.68], *p*<0.001. Left hemispheric location was independently associated with a 28% reduced risk of death, $HR_{0.72}$, 95% CI [0.58–0.89], *p* = 0.003. In Cox adjusted analysis for all prognostic factors, age 60–69 years ($HR_{1.51}$, 95% CI [1.19–1.92], *p* = 0.001) or ≥70 years ($HR_{2.39}$, 95% CI [1.86–3.06],

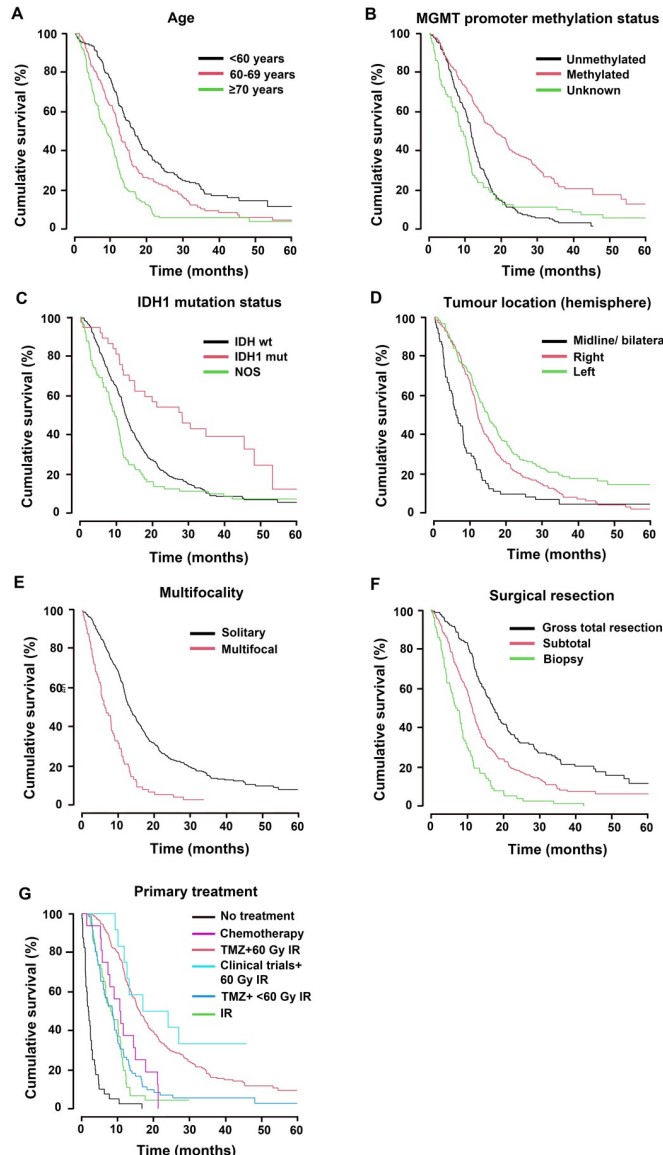

**Fig 2. Prognostic factors and primary treatment associated with patients' overall survival.** Cumulative overall (%) survival time in months after 1-, 2 -and 5-year follow-up for (A) Age, (B) *MGMT* promoter methylation status, (C) *IDH* mutation status, (D) tumour location, (E) tumour multifocality, (F) extent of surgical resection and (G) treatment administered after primary tumour diagnosis. NOS = not otherwise specified.

$p < 0.001$), unmethylated *MGMT* promoter (HR$_{2.09}$, 95% CI [1.66–2.63], $P < 0.001$), right tumour location (HR$_{1.40}$, 95% CI [1.13–1.74], $p = 0.002$), midline or bilateral tumour location (HR$_{1.70}$, 95% CI [1.14–2.55], compared to left tumour location, $p = 0.010$) and multifocality (HR$_{1.57}$, 95% CI [1.04–2.37], $p = 0.034$, Fig 2E) were all independently associated with poor prognosis.

## Extent of tumour resection was associated with patient survival

All patients underwent surgery; the majority (47.3%) had STR, 36.0% GTR, and 16.7% biopsy, Table 1. Compared to GTR (median OS 17.2 months), STR and biopsy both correlated with increased risk of death (median OS 11.4 and 6.8 months, respectively), $p < 0.001$, Fig 2F.

## Treatment administered at primary diagnosis

The treatment administered after primary diagnosis was significantly associated with patients' overall survival, Fig 2G and Table 2. Most patients 60.1% (n = 281) received the Stupp regimen and had a median OS of 16.1 months which was superior to all other groups apart from a selected group of patients included in a clinical trial. In contrast, 15.4% (n = 72) of patients who received temozolomide concomitantly with hypo-fractionated radiotherapy had median OS of 8.5 months. A minority of patients 9.9% (n = 46) who were only administered radiotherapy (IR) survived 8 months, 3.4% (n = 16) received chemotherapy only (median OS 10.6 months), and a selected 2.6% (n = 12) who were enrolled in immunotherapy trials in addition

**Table 2. Treatment characteristics at primary diagnosis and recurrences.**

| Treatment characteristics | N (%) | Survival time from treatment start | | | | Adjusted analyses (Cox) | |
|---|---|---|---|---|---|---|---|
| | | 1 year | 2 years | 5 years | Median OS (months) | Hazard ratio (95% CI) | P-value |
| **Treatment at primary diagnosis** | Total n = 467 (%) | | | | | | |
| TMZ + 60Gy IR | 281 (60.1) | 68.0% | 31.0% | 9.3% | 16.1 | 1 | 1 |
| TMZ + <60Gy IR | 72 (15.4) | 26.4% | 6.9% | 2.8% | 8.5 | 2.60 (1.98, 3.41) | <**0.001** |
| Chemotherapy | 16 (3.4) | 37.5% | 0% | 0% | 10.6 | 2.25 (1.35, 3.75) | **0.002** |
| IR only | 46 (9.9) | 19.6% | 4.4% | 0% | 8.0 | 3.35 (2.40, 4.66) | <**0.001** |
| Clinical trials* | 12 (2.6) | 75% | 50.0% | 0% | 17.0 | 0.66 (0.32, 1.33) | 0.240 |
| No antineoplastic treatment | 40 (8.6) | 2.5% | 0% | 0% | 1.8 | 17.29 (12.00, 24.91) | <**0.001** |
| **Treatment at first tumour recurrence** | Total n = 309 (%) | | | | | | |
| GK/SRS (+/- chemotherapy) | 38 (12.3) | 62.9% | 11.5% | 0% | 14.7 | 0.44 (0.29, 0.67) | <**0.001** |
| IR (+/- chemotherapy) | 5 (1.6) | 20% | 0% | 0% | 6.4 | 0.93 (0.34, 2.54) | 0.890 |
| LAVA | 22 (7.1) | 22.7% | 9.1% | 0% | 6.9 | 0.88 (0.55, 1.41) | 0.603 |
| Chemotherapy | 98 (31.7) | 18.6% | 8.0% | 0% | 5.7 | 1 | 1 |
| Surgery (+/- other treatment) | 48 (15.5) | 24.7% | 11.5% | 0% | 9.3 | 0.62 (0.43, 0.91) | **0.014** |
| Other treatment** | 2 (0.7) | 50.0% | 0% | 0% | 2.1 | 1.00 (0.24, 4.06) | 0.995 |
| No antineoplastic treatment | 96 (31.1) | 2.1% | 0% | 0% | 2.6 | 3.44 (2.54, 4.66) | <**0.001** |
| **Treatment at second tumour recurrence** | Total n = 152 (%) | | | | | | |
| GK/SRS (+/- chemotherapy) | 11 (7.2) | 11.3% | 0% | 0% | 7.1 | 1.10 (0.50, 2.41) | 0.817 |
| IR (+/- chemotherapy) | 7 (4.6) | 14.3% | 0% | 0% | 4.8 | 1.42 (0.60, 3.38) | 0.422 |
| LAVA | 21 (13.8) | 19.1% | 0% | 0% | 6.8 | 1.17 (0.63, 2.16) | 0.621 |
| Chemotherapy | 27 (17.8) | 29.3% | 0% | 0% | 6.2 | 1 | 1 |
| Surgery (+/- other treatment) | 12 (7.9) | 18.2% | 0% | 0% | 6.9 | 0.96 (0.45, 2.05) | 0.909 |
| Other*** | 2 (1.3) | 50.0% | 0% | 0% | 7.1 | 0.39 (0.05, 2.90) | 0.357 |
| No antineoplastic treatment | 72 (47.4) | 4.7% | 0% | 0% | 2.4 | 2.76 (1.67, 4.56) | <**0.001** |
| **Treatment at third tumour recurrence** | Total n = 56 (%) | | | | | | |
| GK/SRS/IR (+/- chemotherapy) | 4 (7.1) | 0% | 0% | 0% | 6.0 | 1 | 1 |
| LAVA | 14 (25.0) | 7.1% | 0% | 0% | 5.5 | 2.51 (0.57, 11.15) | 0.226 |
| Chemotherapy | 2 (3.6) | 0% | 0% | 0% | 6.9 | 2.29 (0.20, 25.65) | 0.503 |
| Surgery (+/- other treatment) | 3 (5.4) | 0% | 0% | 0% | 3.9 | 3.81 (0.62, 23.25) | 0.147 |
| No antineoplastic treatment | 33 (58.9) | 0% | 0% | 0% | 2.7 | 6.75 (1.58, 28.83) | **0.010** |

Significant p-values are highlighted in bold.

*Clinical trials: Immunotherapy or placebo + IR 60 Gy (+/-adjuvant TMZ) (n = 11); clinical trial with dendritic cell vaccination + IR 60Gy (n = 1)

**Other treatment: immunotherapy, private clinic (n = 1); dabrafenib plus trametinib, clinical trial (n = 1)

***Other treatment: Bortezomib and temozolomide, Bortem-17 clinical trial phase Ib (n = 2)

Abbreviations: mOS: median overall survival; CI: confidence interval; GTR: gross total resection; STR: subtotal resection; TMZ: temozolomide; IR: ionizing radiation; Gy: Gray; GK: gamma knife; SRS: stereotactic radiosurgery; LAVA: lomustine, vincristine and bevacizumab

to the 60 Gy IR with or without concomitant TMZ had median OS 17 months, Fig 2G and Table 2. Of patients who did not receive postoperative treatment, 8.6% (n = 40), had a median OS of 1.8 months. In pairwise comparisons using Scheffé's correction for multiple testing, TMZ administered concomitantly with 60 Gy IR was better than IR alone ($p<0.001$), and superior to TMZ given concomitantly with hypo-fractionated IR ($p<0.001$). Standard treatment was also better than chemotherapy only ($p = 0.045$).

To avoid time immortal bias, we set a landmark conditioning on patients reaching 1st, 2nd, and 3rd recurrence. With the primary treatment as a reference, patients experiencing third tumour recurrence had a shorter time to death and had 5x increased mortality risk ($HR_{11.5}$, 95% CI [4.36–30.64], $p<0.0001$), compared to those at first tumour recurrence ($HR_{2.15}$, 95% CI [1.50–3.07], $p<0.0001$) or second recurrence ($HR_{2.25}$, 95% CI [1.08–4.69], $p<0.03$). No treatment administered at the third recurrence had a significant effect on mortality when adjusted for treatment administered at primary diagnosis.

## Treatment administered at first tumour recurrence and survival

Approximately 66.2% (n = 309) of all patients had MRI-confirmed tumour recurrence. The remaining 33.8% had not received antineoplastic treatment at diagnosis (n = 40), had not progressed at the time of censoring, or had no MRI to confirm their progression before death. Most patients with MRI-confirmed first tumour recurrence received antineoplastic therapy, 31.7% (n = 98) received chemotherapy and survived median of 5.7 months after the first recurrence, Table 2. 12.3% (n = 38) of patients who were eligible for gamma knife (GK) or stereotactic radiosurgery (SRS) and the 1.6% (n = 5) of patients who received conventional re-irradiation alone or combined with chemotherapy had 14.7 and 6.4 months median OS respectively, after the first recurrence. 22 (7.1%) patients received the combination of lomustine (100 mg/m$^2$), vincristine (2 mg), and bevacizumab (400 mg day 1 and 15) (LAVA) in a 5-week cycle and had median OS 6.9 months after the first recurrence, while 15.5% (n = 48) who were re-operated only or combined with chemotherapy, had median OS 9.3 months after the first recurrence, Table 2. Patients who did not receive further antineoplastic treatment at first progression, 31.1% (n = 96), had median OS of 2.6 months, Table 2 and S1B Fig. On adjusted analyses using chemotherapy as a comparator, both GK/SRS and surgery alone or combined with chemotherapy had superior survival, $p<0.001$ and $p = 0.014$, respectively. The median time to first progression or death was 8.1 months, whereas the median time from first to second progression or death was 3.5 months.

## Treatment administered at second tumour recurrence and survival

32.5% (n = 152) of all patients had an MRI which confirmed a second tumour recurrence. The remaining 33.7% of whom had been diagnosed with a first recurrence had not received antineoplastic treatment at first recurrence (n = 96), had not progress for the second time at time of censoring, or had no MRI to confirm their progression before death. The majority of them, 47.4% (n = 72), did not receive further treatment and had 2.4 months median OS from the second progression. 17.8% (n = 27) of patients received non-bevacizumab-containing chemotherapy with median OS of 6.2 months, Table 2. Patients treated with LAVA 13.8% (n = 21) had median OS of 6.8 months after the second recurrence, and those who received GK/SRS alone or with chemotherapy 7.2% (n = 11) had median OS of 7.1 months, Table 2 and S1C Fig. After Scheffé's correction for multiple testing, LAVA ($p = 0.042$), and chemotherapy ($p = 0.008$) were more beneficial than no antineoplastic treatment at second recurrence. Median time from the second to third progression or death was 2.1 months.

## Treatment administered at third tumour recurrence and survival

Only 12.0% (56) of all patients were diagnosed with a third tumour recurrence on MRI. The remaining 20.5% of whom had been diagnosed with a second recurrence had not received antineoplastic treatment at second recurrence (n = 72), had not progressed for the third time at time of censoring, or had no MRI to confirm their progression before death. Therapy at third recurrence included LAVA 25% (n = 14), GK/SRS/IR with or without chemotherapy 7.1% (n = 4), chemotherapy not containing bevacizumab 3.6% (n = 2), and re-operation alone or combined with chemotherapy 5.4% (n = 3), Table 2 and S1D Fig. After Scheffé's correction for multiple testing, only LAVA showed an advantage over no treatment at third recurrence (p = 0.043). Treatment upon tumour recurrence differed between the two institutions, Table 3, but was not associated with a significant difference in overall survival. Patients treated at OUH (n = 327) had median OS 12.2 months compared to 11.8 months of patients treated at HUH (n = 140); $HR_{1.145}$ 95% CI [0.93–1.41], p = 0.201, Fig 1B and S1 File.

## Discussion

This is a population-based study of 467 consecutive patients treated for histologically verified GBM at the two largest tertiary referral hospitals in Norway. Median overall survival was 12.1 months with no significant difference between the two hospitals. The prognostic significance of age, *MGMT* promoter status, as well as extent of resection, does not differ from previous studies [12, 13]. Approximately 39% and 13% of patients who harboured tumours with hyper-methylated *MGMT* promoter survived 2 and 5 years, respectively, and confirms the prognostic and predictive value of methylation status [5, 6].

The association between the hemispheric side of tumour location and survival was surprising. Patients with primary tumour in the right hemisphere had poorer outcomes than those with GBM in the left side of the brain. We speculate that tumours localised in the non-dominating hemisphere might grow bigger before being symptomatic, leading to delayed diagnosis [14]. A relationship between neuropsychological test performance and hemispheric tumour site has been described [15] where the left hemisphere tumours were associated with lower scores on verbal tests. Our results are contradictory to another study [16] where left hemisphere tumour was associated with inferior PFS. Localisation of multifocal lesions to the left or right hemisphere had no impact on survival. This may be caused by the detrimental effect of multifocality on prognosis.

*MGMT* promoter methylation status is known to be a positive prognostic and predictive factor for response to therapy in GBM patients, a conclusion corroborated by our findings. Through the revision of the fifth edition of the WHO Classification of Tumours of the Central Nervous System in 2021, the classification of gliomas underwent major changes and the term *IDH*-mutant glioblastoma is no longer applicable [17]. Mutation in the *IDH* genes (*IDH1* or *IDH2)* was previously defined to occur in 5–10% of all GBM and was associated with secondary GBM, younger age, and better outcome [18, 19]. According to the previous classification, GBM harbouring *IDH* mutation was present in 7.9% of patients in the herein presented cohort, and had a significant positive association with survival on unadjusted analyses, but not on adjusted analyses. However, on adjusted analyses for the primary GBM cohort only, we found that *IDH* mutation had a significant positive association with survival (p = 0.006). According to the new WHO classification, those tumours are now defined as IDH mutated astrocytoma grade IV, which underscores the improved survival reported in this study. However, the data were collected and analysed before the new classification was published and the patients received the glioblastoma treatment so the authors chose not to exclude this group of patients from the final analysis. For 80 patients (17.1%) *MGMT* promoter methylation status

**Table 3. Patient, tumour and treatment characteristics at two independent institutions.**

| Tumour and treatment characteristics | OUH | HUH |
|---|---|---|
| **Age (years)** | Total n = 327 (%) | Total n = 140 (%) |
| <60 | 134 (41.0) | 53 (37.9) |
| 60–69 | 96 (29.4) | 48 (34.3) |
| ≥70 | 97 (29.7) | 39 (27.9) |
| **Tumour location** | Total n = 327 (%) | Total n = 140 (%) |
| Right side | 144 (44.0) | 68 (48.6) |
| Left side | 134 (41.0) | 49 (35.0) |
| Midline/bilateral | 49 (15.0) | 23 (16.4) |
| **Multifocality** | Total n = 327 (%) | Total n = 140 (%) |
| Solitary | 273 (83.5) | 118 (84.3) |
| Multifocal | 54 (16.5) | 22 (15.7) |
| ***MGMT* promoter status** | Total n = 327 (%) | Total n = 140 (%) |
| Hypermethylated | 108 (33.0) | 75 (53.6) |
| Unmethylated | 141 (43.1) | 63 (45.0) |
| Unknown | 78 (23.9) | 2 (1.4) |
| ***IDH* status** | Total n = 327 (%) | Total n = 140 (%) |
| Mutated | 26 (8.0) | 11 (7.9) |
| Wild type | 227 (69.4) | 122 (87.1) |
| Unknown | 74 (22.6) | 7 (5.0) |
| **Extent of surgical resection at primary diagnosis** | Total n = 327 (%) | Total n = 140 (%) |
| GTR | 118 (36.1) | 50 (35.7) |
| STR | 153 (46.8) | 68 (48.6) |
| Biopsy | 56 (17.1) | 22 (15.7) |
| **Treatment at primary diagnosis** | Total n = 327 (%) | Total n = 140 (%) |
| TMZ + 60Gy IR | 201 (61.5) | 80 (57.1) |
| TMZ + <60Gy IR | 46 (14.1) | 26 (18.6) |
| Chemotherapy | 14 (4.3) | 2 (1.4) |
| IR only | 25 (7.7) | 21 (15) |
| Clinical trials* | 12 (3.7) | 0 |
| No antineoplastic treatment | 29 (8.9) | 11 (7.9) |
| **Treatment at first tumour recurrence** | Total n = 210 (%) | Total n = 99 (%) |
| GK/SRS (+/- chemotherapy) | 8 (3.8) | 30 (30.3) |
| IR (+/- chemotherapy) | 4 (1.9) | 1 (1.0) |
| LAVA** | 4 (1.9) | 18 (18.2) |
| Chemotherapy | 91 (43.3) | 7 (7.1) |
| Surgery (+/- other treatment) | 33 (15.7) | 15 (15.2) |
| Other treatment*** | 1 (0.5) | 1 (1.0) |
| No antineoplastic treatment | 69 (32.9) | 27 (27.3) |
| **Treatment at second tumour recurrence** | Total n = 91 (%) | Total n = 61 (%) |
| GK/SRS (+/- chemotherapy) | 1 (1.1) | 10 (16.4) |
| IR (+/- chemoterapy) | 6 (6.6) | 1 (1.6) |
| LAVA** | 1 (1.1) | 20 (32.8) |
| Chemotherapy | 22 (24.2) | 5 (8.2) |
| Surgery (+/- other treatment) | 6 (6.6) | 6 (9.8) |
| Other**** | 1 (1.1) | 1 (1.6) |
| No antineoplastic treatment | 54 (59.3) | 18 (29.5) |
| **Treatment at third tumour recurrence** | Total n = 20 (%) | Total n = 36 (%) |

*(Continued)*

**Table 3.** (Continued)

| Tumour and treatment characteristics | OUH | HUH |
|---|---|---|
| GK/SRS/IR (+/- chemotherapy) | 2 (10.0) | 2 (5.6) |
| LAVA** | 0 (0) | 14 (38.9) |
| Chemotherapy | 2 (10.0) | 0 (0) |
| Surgery (+/- other treatment) | 0 (0) | 3 (8.3) |
| No antineoplastic treatment | 16 (80.0) | 17 (47.2) |

*Clinical trials: Immunotherapy or placebo + IR 60 Gy (+/-adjuvant TMZ) (n = 11); clinical trial with dendritic cell vaccination + IR 60Gy (n = 1)

**LAVA registered at OUH is either bevacizumab in combination with lomustine, bevacizumab monotherapy or LAVA administered at HUH.

***Other treatment: immunotherapy, private clinic (n = 1)(HUH); dabrafenib plus trametinib, clinical trial (n = 1) (OUH)

****Other treatment: Bortezomib and temozolomide, Bortem-17 clinical trial phase Ib (n = 2)

Abbreviations: OUH: Oslo University Hospital; HUH: Haukeland University Hospital; *IDH*: isocitrate dehydrogenase; *MGMT*: O$^6$ methylguanine-DNA methyltransferase; GTR: gross total resection; STR: subtotal resection; TMZ: temozolomide; IR: ionizing radiation; Gy: Gray; GK: gamma knife; SRS: stereotactic radiosurgery; LAVA: lomustine, vincristine and bevacizumab

was unknown. As these patients had an overall survival inferior to that of patients with methylated *MGMT*, we presume that *MGMT* status was unmethylated in most of these patients' GBM's [5, 6]. Another possible explanation is that as many as 42.5% of these patients underwent biopsy only, in contrast to 13.7% of patients with unmethylated *MGMT*. This factor may also explain the inferior outcome of patients with unknown *IDH* mutation status, of which 40.7% underwent biopsy only compared to 12.0% of patients with *IDH* wild type GBM.

The association of the extent of surgical resection on survival is in concordance with previous studies. STR was associated with better survival than biopsy only [20, 21], which underscores the significance of extent of resection even if GTR is not possible. Elderly patients and patients with poor performance status received hypo-fractionated radiotherapy [22–24]. They had inferior survival compared to those treated with the Stupp regimen [5]. Poor performance status and advanced age are known factors contributing to shorter survival of patients with high-grade glioma [25, 26]. However, 27 (19.9%) patients of 70 years and older who were fit enough to receive the Stupp regimen had a median OS of 14.3 months. This indicates the need for individualised approach for elderly GBM patients.

The median time to the first progression was 8.1 months, which is longer than reported in the literature [5]. This may be caused by the above-mentioned selection of patients that survived to the first control MRI. Only 66.2% of patients had MRI-confirmed tumour relapse. The majority of patients were treated with non-bevacizumab containing chemotherapy at first relapse and the treatment of choice was re-challenge with temozolomide, the latter providing that the patients did not progress under the adjuvant treatment with temozolomide. This unfortunately generates a selection bias that is difficult to avoid in the real-life retrospective analysis. At first recurrence (n = 309), 15.5% of patients were treated with a second surgery and 12.3% received GK or SRS alone or combined with chemotherapy. Both treatments were associated with superior survival on adjusted analyses when compared to chemotherapy alone ($p<0.001$ and $p = 0.014$, respectively). GK/SRS is regarded as a time-efficient salvage treatment similar to debulking surgery for small lesions that enables postponing new systemic therapy until later progressions. The superior OS of patients treated with GK/SRS at first relapse may be explained by a small volume of relapsed lesions and deferring chemotherapy and secondary

resistance to alkylating agents [27]. The GK/SRS is applied both at early and delayed recurrence. Thus, even patients with relapse during adjuvant chemotherapy are considered for this treatment. We postulate, that the time intevals between control MRIs are important to detect smaller lesions available for volume-limited irradiation and may facilitate deferring new systemic therapy. The superior OS of patients treated with GK/SRS at first relapse indicates that the detected recurrent tumour was small enough to qualify for this treatment, the lesion was uni- or bi-focal, the new line of systemic therapy was deferred and the MRI follow-up frequency allowed detection of tumour progression when the patients were fit enough to receive further treatment. However, fractionated RT is usually avoided during the first year following the initial treatment. Generally, a second course of fractionated RT is postponed and often applied as the last treatment option in patients that are not available for systemic therapy, GK/SRS, and/or repeated surgery. The sample size is the limiting factor for a separate analysis of this patient group.

LAVA is a combination of lomustine, bevacizumab, and vincristine, a therapy regimen formulated and used at HUH. Patients treated with LAVA at third recurrence did better than those receiving no treatment at all, but the selection bias related to performance status must be considered. LAVA is applied only as the last line of treatment in selected patients with relapsed tumours not available for GK/SRS or re-challenge with TMZ. Thus, the selection bias may also explain the inferior survival of patients treated with LAVA at first relapse indicating early relapse and bigger lesions, precluding GK/SRS treatment.

Bevacizumab in combination with lomustine prolonged PFS, but not OS in relapsed GBM in a phase III trial [28, 29]. Although bevacizumab administered with or without chemotherapy did not meet expectations generated by the BELOB trial [30], it is frequently applied in clinical practice due to the paucity of other therapeutic options. In the analysed cohort, treatment with LAVA was associated with a slightly reduced risk of death.

The choice of subsequent lines of treatment differed between the two institutions, reflecting the lack of standard recommended treatment for recurrent GBM. Moreover, the choice of therapy depends on both patient- and tumour-related factors. The major differences between the two institutions were the use of bevacizumab and widely used GK at one of the centres, and participation in international clinical trials at the other. However, there were no significant differences in overall survival.

Our study is limited by patient selection, small sample sizes, and heterogeneous treatment groups, especially at second and third recurrences. We conclude that despite a heterogeneous approach to the therapy of recurrent GBM, there are no major differences in median OS. Patients who are deemed fit to receive antineoplastic treatment at recurrence may benefit from individually adjusted therapy. However, novel treatment options for recurrent GBM, especially addressing tumours with unmethylated *MGMT* and resistance to temozolomide, are urgently needed.

## Supporting information

**S1 File. Methods, results, references, and figure legends.**
(DOCX)

**S1 Fig. Effect of treatment administered at subsequent tumour recurrence on patients' overall survival.** (A) Cumulative overall (%) survival time in months and 95% confidence intervals after 1 year, Cumulative overall (%) survival time from diagnosis in months after 1-, 2 -and 5-year follow-up for (B) treatment administered after first tumour recurrence, (C) treatment administered after second tumour recurrence, and (D) treatment administered after third tumour recurrence. LAVA: lomustine, bevacizumab and vincristine; TMZ:

temozolomide; IR: ionizing radiation; Gy: gray; SRS: stereotactic radiosurgery; GK: gamma knife; Other: GK/SRS/IR (+/- chemotherapy), Chemotherapy, Surgery (+/- chemotherapy). (TIF)

## Author Contributions

**Conceptualization:** Martha Chekenya, Dorota Goplen.

**Data curation:** Hanne Blakstad, Jorunn Brekke, Mohummad Aminur Rahman, Victoria Smith Arnesen, Petter Brandal, Martha Chekenya.

**Formal analysis:** Hanne Blakstad, Stein Atle Lie, Martha Chekenya.

**Funding acquisition:** Martha Chekenya, Dorota Goplen.

**Investigation:** Martha Chekenya.

**Methodology:** Stein Atle Lie, Martha Chekenya.

**Project administration:** Mohummad Aminur Rahman, Martha Chekenya.

**Resources:** Martha Chekenya.

**Software:** Stein Atle Lie.

**Supervision:** Martha Chekenya.

**Validation:** Martha Chekenya.

**Visualization:** Mohummad Aminur Rahman.

**Writing – original draft:** Hanne Blakstad, Martha Chekenya, Dorota Goplen.

**Writing – review & editing:** Hanne Blakstad, Jorunn Brekke, Mohummad Aminur Rahman, Victoria Smith Arnesen, Hrvoje Miletic, Petter Brandal, Stein Atle Lie, Martha Chekenya, Dorota Goplen.

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
