## [Decision Letter · Decision Letter 0]

4 Oct 2022

PONE-D-22-23881­­­Survival in a consecutive series of 467 glioblastoma patients: association with prognostic factors and treatment at recurrence at two independent institutionsPLOS ONE

Dear Dr. Chekenya,

Thank you for submitting your manuscript to PLOS ONE. After careful consideration, we feel that it has merit but does not fully meet PLOS ONE’s publication criteria as it currently stands. Therefore, we invite you to submit a revised version of the manuscript that addresses the points raised during the review process.

We look forward to receiving your revised manuscript.

Kind regards,

Alvan Ukachukwu, MD, MSc.GH

Academic Editor

PLOS ONE

Journal Requirements:

"We are grateful to the Norwegian Cancer Society (grant # 190170) and KLINBEFORSK for supporting our research. "

"1. MC 

Grant # 190170 

Norwegian Cancer Society 

https://kreftforeningen.no/

Sponsors or funders did not play any role in the study design, data collection and analysis, decision to publish, or preparation of the manuscript

2. MC and DG

Program for klinisk behandlingsforskning - KLINBEFORSK

https://kliniskforskning.rhf-forsk.org/

Sponsors or funders did not play any role in the study design, data collection and analysis, decision to publish, or preparation of the manuscript"

We will update your Data Availability statement on your behalf to reflect the information you provide.Please include your full ethics statement in the ‘Methods’ section of your manuscript file. In your statement, please include the full name of the IRB or ethics committee who approved or waived your study, as well as whether or not you obtained informed written or verbal consent. If consent was waived for your study, please include this information in your statement as well. 

Additional Editor Comments:

Kindly respond to the reviewers comments, especially regarding treatment strategy at initial occurrence and at recurrence, and the risk of bias.

Reviewers' comments:

Reviewer's Responses to Questions

**Comments to the Author**

1. Is the manuscript technically sound, and do the data support the conclusions?

Reviewer #1: Yes

Reviewer #2: Partly

2. Has the statistical analysis been performed appropriately and rigorously? 

Reviewer #1: Yes

Reviewer #2: Yes

3. Have the authors made all data underlying the findings in their manuscript fully available?

Reviewer #1: Yes

Reviewer #2: No

4. Is the manuscript presented in an intelligible fashion and written in standard English?

Reviewer #1: Yes

Reviewer #2: Yes

5. Review Comments to the Author

Reviewer #1: The reviewed manuscript Survival in a consecutive series of 467 glioblastoma patients: association with prognostic factors and treatment at recurrence at two independent institutions is well structured and neatly presented retrospective study that sheds light on outcomes in treatment of recurrent glioblastoma, considering the intrinsic biological, pathological and clinical features of the disease, as well as primary therapeutic modalities.

The manuscript is delivered in a clear and precise, easily comprehensible manner. The methodology is sufficiently detailed, the results are presented in illustrative tables and figures, and conclusions are adequate and well elaborated.

The problem analyzed in the present study is of major clinical relevance. There is no established treatment for GBM recurrence, thus the results of the present study may be significant in treatment decision making for patients and may contribute to the improvement of patients' outcomes. Given that the research includes two centers with, to some extent, different treatment approaches, the results of the study may represent indicative guidepost for the treatment of recurrent GBM depending on the available resources at the institution.

It was a pleasure to read this manuscript. I find this article acceptable to be published in present form.

Reviewer #2: Thank you for the opportunity to review your work. The authors investigated physicians’ treatment choice at recurrence and prognostic and predictive factors for survival in GBM patients from Norway’s two largest regional hospitals.

1) there are a few grammatical errors here and there that may require a review

2) the prognosis factors associated with GBM has been investigated in published studies, no additional information could be available from present study.

3) The treatment strategy at second recurrence would be significantly impacted by first-line treatment. For example, 50% of GBM recurred in one year, if GBM received first-line concurrent chemotherapy, most radiation oncologists would recommend to other treatment but not for radiotherapy due to its high-risk of toxicities. As a result, the analysis of treatment strategy at recurrence is obviously biased in the present study.

6. PLOS authors have the option to publish the peer review history of their article (what does this mean?). If published, this will include your full peer review and any attached files.

Reviewer #1: **Yes: **Miljan Krstic, MD PhD

Reviewer #2: No

---

## [Author Response · Author response to Decision Letter 0]

20 Nov 2022

Authors: Thank you. We have formatted the manuscript as requested.

Authors: Thank you. The information provided in the “funding information” section is correct.

"We are grateful to the Norwegian Cancer Society (grant # 190170) and KLINBEFORSK for supporting our research. "

"1. MC 

Grant # 190170 

Norwegian Cancer Society 

https://kreftforeningen.no/

Sponsors or funders did not play any role in the study design, data collection and analysis, decision to publish, or preparation of the manuscript

2. MC and DG

Program for klinisk behandlingsforskning - KLINBEFORSK

https://kliniskforskning.rhf-forsk.org/

Sponsors or funders did not play any role in the study design, data collection and analysis, decision to publish, or preparation of the manuscript"

Authors: we have now removed the acknowledgement of funding from the manuscript. We confirm that the information provided in the “funding information” section is correct.

Authors: Data cannot be shared publicly because of this is sensitive patient data. Due to these restrictions on public sharing for participant privacy, Data approved by the Ethics Committee (approval number 2017/2084) will be available from the corresponding authors upon request. 

We will update your Data Availability statement on your behalf to reflect the information you provide.Please include your full ethics statement in the ‘Methods’ section of your manuscript file. In your statement, please include the full name of the IRB or ethics committee who approved or waived your study, as well as whether or not you obtained informed written or verbal consent. If consent was waived for your study, please include this information in your statement as well. 

Authors: This information that was previously in the supplementary information, is now added to the main manuscript. 

Full name of the IRB or ethics committee and contact details:

Regional Ethics Committee, REK vest

Armauer Hansens House (AHH),

Tverrfløy Nord, 2 floor. Room

281. Haukelandsveien 28, 5025 Bergen, Norway

Telefon: 55975000

E-post: rek-vest@uib.no

Web: http://helseforskning.etikkom.no/

 Kindly address all mail and e-mails to the Regional Ethics Committee, REK vest, not to individual staff

Non-author institutional contact person:

Berit Bølge Tysnes, PhD

Department of Biomedicine

University of Bergen

Jonas Lies vei 91

5009 Bergen, Norway

Email: Berit.Tysnes@uib.no

Phone: +4790778791 / +4755586093

Authors: Regional Committee for Medical and Research Ethics for Western Norway (REC West) approved the retrospective patient identification and collection of clinicopathological data (2017/2084). Exemption from the need to obtain informed consent from included patients, including the few surviving patients at the time of data collection (n=9 and n=43; HUH and OUH, respectively), was granted by REC West.

Authors: References have been reviewed and corrected.

Additional Editor Comments:

Kindly respond to the reviewers comments, especially regarding treatment strategy at initial occurrence and at recurrence, and the risk of bias.

Authors: To analyse the impact of the previous treatment on the reccurent treatments, analysis using time-dependent covariates were performed. Furthermore, to avoid time immortal bias, we set a landmark conditioning on patients reaching 1st, 2nd and 3rd recurrence.

Reviewers' comments:

 Comments to the Author

5. Review Comments to the Author

Reviewer #1: The reviewed manuscript Survival in a consecutive series of 467 glioblastoma patients: association with prognostic factors and treatment at recurrence at two independent institutions is well structured and neatly presented retrospective study that sheds light on outcomes in treatment of recurrent glioblastoma, considering the intrinsic biological, pathological and clinical features of the disease, as well as primary therapeutic modalities.

The manuscript is delivered in a clear and precise, easily comprehensible manner. The methodology is sufficiently detailed, the results are presented in illustrative tables and figures, and conclusions are adequate and well elaborated.

The problem analyzed in the present study is of major clinical relevance. There is no established treatment for GBM recurrence, thus the results of the present study may be significant in treatment decision making for patients and may contribute to the improvement of patients' outcomes. Given that the research includes two centers with, to some extent, different treatment approaches, the results of the study may represent indicative guidepost for the treatment of recurrent GBM depending on the available resources at the institution.

It was a pleasure to read this manuscript. I find this article acceptable to be published in present form.

Authors: We are grateful to the reviewer for constructive feedback that greatly improved the quality of the work.

Reviewer #2: Thank you for the opportunity to review your work. The authors investigated physicians’ treatment choice at recurrence and prognostic and predictive factors for survival in GBM patients from Norway’s two largest regional hospitals.

1) there are a few grammatical errors here and there that may require a review

Authors: Thank you, we have now thoroughly proof read the manuscript and corrected all grammatical errors.

2) the prognosis factors associated with GBM has been investigated in published studies, no additional information could be available from present study.

Authors: This is indeed true. We agree with the reviewer. We controlled for these established biomarkers in our study in order to provide quality assurance of our patient cohort, ensuring that the findings are as expected from the literature.

3) The treatment strategy at second recurrence would be significantly impacted by first-line treatment. For example, 50% of GBM recurred in one year, if GBM received first-line concurrent chemotherapy, most radiation oncologists would recommend to other treatment but not for radiotherapy due to its high-risk of toxicities. As a result, the analysis of treatment strategy at recurrence is obviously biased in the present study.

Authors: We agree with the reviewer that the treatment at the second recurrence is impacted by the first line treatment. We did attempt to control the inherent bias with such retrospective analyses. To analyse the impact of the previous treatment on the recurrent treatments, analysis using time-dependent covariates were performed. Furthermore, to avoid time immortal bias, we set a landmark conditioning on patients reaching 1st, 2nd and 3rd recurrence. We totally agree that the re-irradiation is not recommended during the first year following postoperative radiotherapy. However, those limitations do not apply to the stereotactic radiosurgery/GK, which due to the limited volume of irradiated tissue and little toxicity, is often the treatment of choice for small contrast enhancing relapses. The selection of chemotherapy regimen at relapse/progression also depends on timing of treatment failure/relapse. Progressive disease during adjuvant therapy precludes use of temozolomide, indicating the primary resistance. We have now included this statement in the discussion. 

Those real-life data elucidating the available treatment options may be useful for clinicians to select treatment for recurrent glioblastoma patients in the paucity of positive clinical trials. We propose that, however limited, the available therapies may be considered for individual approach.

---

## [Editor Report · Decision Letter 1]

17 Jan 2023

­­­Survival in a consecutive series of 467 glioblastoma patients: association with prognostic factors and treatment at recurrence at two independent institutions

PONE-D-22-23881R1

Dear Dr. Chekenya,

We’re pleased to inform you that your manuscript has been judged scientifically suitable for publication and will be formally accepted for publication once it meets all outstanding technical requirements.

Kind regards,

Alvan Ukachukwu, MD, MSc.GH

Academic Editor

PLOS ONE